# Optimization of two-passenger ride-pooling orders based on ST-GNN and path optimization

**Xue Xing** [iD]*, **Yuqi Peng, Le Wan, Fahui Luo**

Jilin University of Chemical Technology, The School of Information and Control Engineering, Jilin, China

* xingx@jlict.edu.cn

## Abstract

Urban dynamic ride-pooling faces significant challenges in achieving efficient real-time order matching and path planning, primarily due to the complex spatio-temporal coupling of passenger demand and traffic conditions. Traditional algorithms often struggle to dynamically integrate these features and adapt to multi-objective optimization under real-world constraints. To address these limitations, this study proposes a novel dual-optimization framework that synergizes a Spatio-Temporal Graph Neural Network (ST-GNN) with a multi-objective path planning algorithm. Our approach begins by constructing a demand-adaptive urban spatial structure using Voronoi polygons. A spatio-temporal graph is then built upon this structure, where a graph neural network model, incorporating multi-head attention and Transformer mechanisms, is employed to learn node embeddings that capture complex urban dynamics. These embeddings inform the matching of suitable ride-pooling pairs and guide an improved Dijkstra algorithm to generate optimal paths that co-optimize travel distance, passenger detour, and carbon emissions while strictly adhering to passenger time windows. Validated on a large-scale real-world dataset from Chengdu (Didi Chuxing), our method achieves a matching success rate of 86.6%, reduces carbon emissions by 0.34 kg $CO_2$ per order on average, and maintains a low average detour rate of 0.1202. The results demonstrate that the proposed model enhances spatio-temporal collaboration in complex scenarios and offers a practical and efficient solution for the intelligent upgrade of shared mobility systems, contributing to optimized urban traffic resources and low-carbon travel practices.

## Introduction

With the acceleration of urbanization and the aggravation of traffic congestion, shared travel mode has gradually become an important part of the urban transportation system [1]. As the core carrier of shared travel, the dynamic ride-sharing technology of e-hailing is considered a promising approach for improving capacity efficiency and reducing carbon emissions [2]. However, existing technologies face two core challenges. First, traditional algorithms are difficult to effectively capture the

**Data availability statement:** Publicly accessible dataset hosted on Kaggle: https://www.kaggle.com/datasets/yunibibi/dataset.

**Funding:** This work was supported by the Jilin Province Science and Technology Development Plan Funded Project [Grant Number YDZJ202301ZYTS291] AND the Industrialization Cultivation Project of the Jilin Provincial Department of Education [Grant Number JJKH20230306CY].

**Competing interests:** The authors have declared that no competing interests exist.

complex interaction between dynamic traffic states and passenger demand due to the insufficient fusion of spatio-temporal features. Second, the dynamic adaptability is weak, and the optimization ability of existing methods is limited under real-time traffic constraints. Especially in short-distance order scenarios in commercial dense areas, how to achieve efficient matching and path planning has become a technical bottleneck to be broken through.

In recent years, Graph Neural Networks (GNNs) have shown unique advantages in the field of intelligent transportation [3]. It models the non-Euclidean characteristics of the transportation network through the graph structure, which can effectively capture the spatial dependence and dynamic evolution law between nodes. These techniques have been successfully applied to various traffic prediction tasks, such as traffic flow forecasting [4] and bike-sharing demand prediction around metro stations [5]. Combined with the path optimization algorithm, GNN has the potential to offer an alternative solution for the ride-pooling problem. However, the existing research still has shortcomings in the coupling modeling of regional division characteristics and dynamic order distribution, and the real-time guarantee of multi-objective optimization. Now, it is necessary to explore more refined spatio-temporal perception and collaborative optimization methods.

The research on dynamic ride-pooling routing optimization has made significant progress worldwide, covering many aspects such as algorithm design, model construction, practical application and technical challenges. Foreign research focuses on algorithm innovation and multi-objective optimization. Amores et al. (2021) proposed a "recoverable path planning" model to improve navigation robustness by introducing a fault-tolerant mechanism, but did not consider the multi-user collaboration problem in dynamic ridesharing scenarios [6]. Li et al. (2020) studied the dynamic route recommendation problem to optimize user experience by continuously reporting alternative routes, but their model did not involve carbon emissions and vehicle capacity constraints [7]. For example, the Enhanced Spatio-temporal Attention Graph Neural Network (RSTAG) proposed by Zhou et al. (2020) performs well in traffic flow prediction, but its application scenario is limited to prediction rather than path optimization [4]. In order to improve the solution efficiency and global search ability, Lu et al. (2023) designed an improved PSO-NSGA-II algorithm that combines the memory characteristics of particle swarm optimization and the advantages of non-dominated sorting genetic algorithm. Through the case analysis from Xi 'an to Xianyang, the effectiveness of the model and algorithm is verified. The results show that the proposed algorithm has a significant improvement in global search ability and convergence speed compared with the standard NSGA-II [8].

Domestic research relies more on traditional optimization algorithms. Wu Yuailin et al. (2020) built a taxi pooling model based on trajectory similarity, and improved matching efficiency through a two-stage algorithm (k-medoids clustering + ant colony optimization), but did not consider the influence of real-time traffic state on the path [9]. The Graph Convolutional Network (GCN) short-term traffic flow prediction method proposed by Chen Danlei et al. (2021) has certain innovation in spatio-temporal feature extraction, but its model is not directly related to carpooling path

planning [10]. In addition, Z Xia et al. (2024) 's Dynamic Spatio-temporal graph convolutional Recurrent Network (DST-GRNN) has made progress in traffic flow prediction, but it also lacks the exploration of multi-user collaborative optimization [5]. Xue Shouqiang et al. (2021) constructed a multi-objective mixed integer linear programming model for ridesharing with the goal of maximizing passenger satisfaction and minimizing travel time [11].

However, previous efforts in ride-pooling optimization often treat spatio-temporal feature extraction, dynamic adaptability, and multi-objective planning as sequential or separate challenges, leading to insufficient feature fusion and limited performance in real-time scenarios. This gap motivates our work. We propose a dual-optimization framework that integrates a Spatio-Temporal Graph Neural Network (ST-GNN) with path optimization. Our approach is distinct in its tight coupling of a demand-aware spatial topology (Voronoi polygons) with a deep learning model (GAT and Transformer) within a unified graph structure. This integrated design enables the simultaneous capture of complex spatio-temporal dependencies and the direct generation of optimal paths via an improved Dijkstra algorithm, thereby effectively addressing the conflicts between matching efficiency, passenger experience, and environmental goals.

## Description of the two-order pickup routing problem

### Problem definition

Ride-hailing ride-pooling mainly comes in two forms: two-passenger ride-pooling and multiple ride-pooling. The core lies in integrating orders with similar routes or time Windows to enhance vehicle utilization, reduce empty running rates, and thereby lower operating costs and carbon emissions. Two-Passenger Ride-Pooling ($R_2$) refers to a ride-hailing vehicle carrying two passengers on the same trip, forming four specific pick-up and drop-off points. This model achieves a good balance between enhancing resource utilization efficiency and controlling algorithm complexity. It can effectively integrate transportation capacity while maintaining a relatively controllable complexity of order matching and path planning. It is particularly suitable for short-distance order scenarios such as commercial dense areas. In contrast,Multi-Passenger Ride-Pooling ($R_x$, $x>2$) refers to a vehicle serving more than two passengers in the same trip, that is, the total number of orders x belongs to the set X (X = 3, 4,...) (xmax). Although the multi-combination mode can further enhance the passenger carrying capacity and vehicle utilization efficiency of a single trip, especially suitable for areas with high travel demand or peak hours, as the number of orders increases and the number of pick-up and drop-off points involved also increases, the complexity of route planning combinations (such as pick-up and drop-off sequences) and the difficulty of order matching also significantly increase Higher requirements have been put forward for the real-time performance, optimization ability of the algorithm and the response speed of the scheduling system.

The problem of optimizing ride-pooling routes for online car-hailing services is a series of methods and strategies that scientifically plan and dynamically adjust the driving routes of online car-hailing vehicles through technical means to maximize the satisfaction of passengers' travel demands, reduce operating costs, increase vehicle utilization rates, and minimize the waste of transportation resources [12]. The core lies in matching passengers' starting and ending points, travel time and other demands through algorithm models, dynamically generating the optimal driving path, while balancing multiple goals such as passengers' travel time and drivers' detour costs.

This study focuses on the matching problem between two orders based on path optimization in the urban shared ride-hailing scenario. By constructing a spatio-temporal graph model that integrates the topological features of Voronoi polygons and dynamic order characteristics, and combining it with a multi-dimensional attention mechanism, the coupled modeling of static attributes of the road network (such as node connection relationship and road length) and dynamic attributes (such as node order volume and real-time road network speed) is achieved. This method aims to break through the limitations of traditional algorithms in dynamic adaptability and spatio-temporal feature fusion, effectively address the spatio-temporal coupling constraints in complex urban traffic scenarios (such as passenger time Windows), and minimize carbon emissions and detour rates under the condition of meeting the constraints.

**Problem hypothesis**

1. Structural stability of the road network: the topological relationship between nodes (regional centers of Theisons) and edges (inter-regional connected roads) of the urban road network remains stable during the planning period, and only the edge weights (travel time/distance) are dynamically adjusted according to the real-time traffic state.
2. Order information integrity: The starting and ending points, estimated departure time, latest pickup time (time window) and number of passengers of each order are known and accurate.
3. Passenger waiting willingness: Passengers can accept detour or waiting time within a certain range, but the actual pickup time shall not exceed the latest allowed time window set by them.

**Methodology overview**

To systematically address the defined problem, this study develops an integrated methodology that bridges spatio-temporal perception with real-time path optimization. The overall workflow is structured around four core components. The process begins by constructing a demand-responsive representation of the city through Voronoi-based spatial partitioning, forming a graph where nodes are regions and edges are road connections. Upon this graph, a Spatio-Temporal Graph Neural Network (ST-GNN) is constructed, which synergizes Graph Attention Networks for spatial modeling and Transformer encoders for temporal dynamics to generate comprehensive node embeddings. These embeddings subsequently enable a two-fold function: they inform the matching probability between orders and dynamically adjust graph edge weights to reflect real-time conditions. Ultimately, for each matched pair, an improved Dijkstra algorithm utilizes these refined weights to generate the optimal joint path, effectively co-optimizing for carbon efficiency and low detour rates under all specified constraints. The following sections detail each of these components.

## Dual-order pick-up and drop-off route optimization model based on ST-GNN

### Area division of road network

To address the problem of insufficient representation of ride-hailing demand space by traditional grid-based or administrative boundary divisions, this paper employs Voronoi polygons to generate non-uniform regions, offering significant advantages for ride-pooling applications. Unlike uniform grid partitions that maintain fixed cell sizes regardless of demand density, or administrative boundaries that are static and often misaligned with travel patterns, Voronoi polygons adaptively partition the urban space based on actual demand distribution. This approach generates smaller regions in high-demand areas (e.g., city centers) where higher spatial resolution is needed for precise matching, and larger regions in low-demand suburban areas, thus providing a mathematically robust framework for spatial division based on demand density.

In this framework, each cluster center of ride-pooling orders serves as a generator (or seed point). The region for each seed is defined as the set of all locations in the plane that are closer to that seed than to any other. The boundaries of these regions are formed by the perpendicular bisectors of the lines connecting adjacent seed points, creating a convex polygon for each seed. Two key mathematical properties govern this construction: 1) the Adjacency Rule, where two regions are adjacent if and only if their seed points are adjacent in the Delaunay triangulation (the dual graph of the Voronoi diagram), and 2) the Edge Prohibition Rule, which ensures that no edges of the Voronoi polygon cross the perpendicular bisectors, thereby guaranteeing that each polygon edge is a segment of the bisector between two seeds and that all regions are contiguous and non-overlapping.

This demand-adaptive partitioning is particularly suitable for ride-pooling applications, as it naturally aligns with the heterogeneous distribution of travel demand across urban spaces. Empirical studies show that online ride-hailing orders show high-density aggregation in commercial dense areas, while they show scattered distribution characteristics in urban peripheral areas [13]. Define road network connections as edges of the graph. The edge weights are initialized to the historical average travel time and updated dynamically through the vehicle trajectory data. Node features include static attributes such as POI distribution and dynamic attributes such as real-time order quantity, while edge features include

information such as road grade and road number. The graph structure integrates infrastructure and traffic states and provides a data basis for subsequent spatial modeling and path planning.

## The ST-GNN model framework

In this study, Spatio-Temporal Graph Neural Network (ST-GNN), a graph neural network framework that integrates spatio-temporal perception, is used to model the dynamic changes in urban transportation networks, and realize efficient matching and path planning of online ride-pooling orders. As shown in Fig 1.

The spatial modeling adopts two-layer multi-head Graph Attention Network (GAT), which aggregates neighborhood information by calculating attention coefficients between nodes and captures the functional association between different regions. The first GAT layer transforms 5-dimensional static node features to 64-dimensional representations, while the second GAT layer maintains 64-dimensional features with 2 attention heads each, dynamically adjusting the importance of neighbor information according to node feature similarity. The spatial aggregation produces 128-dimensional node representations through concatenation of the 2 attention heads.

For temporal modeling, Transformer encoder layers are employed to capture long-range dependencies and dynamic patterns across multiple time slices. The encoder processes sequences of spatially-aggregated node features over a sliding time window (default size = 3), utilizing multi-head self-attention mechanisms to model complex temporal relationships. The Transformer architecture includes positional encoding to preserve temporal order information and layer normalization for training stability.

The feature fusion employs element-wise addition of spatial and temporal representations:

$$H_{\text{final}} = H_{\text{spatial}} + H_{\text{temporal}} \in \mathbb{R}^{128}$$

where both $H_{\text{spatial}}$ and $H_{\text{temporal}}$ are 128-dimensional vectors. This 128-dimensional spatio-temporal embedding vector is then used for subsequent path planning and order matching tasks.

The model processes dynamic node features (2-dimensional: departures and arrivals) and edge features (2-dimensional: average travel time and standard deviation) through dedicated encoding layers before spatial and temporal aggregation. The final output provides a comprehensive 128-dimensional representation for each node in the graph, capturing both static infrastructure characteristics and dynamic traffic patterns.

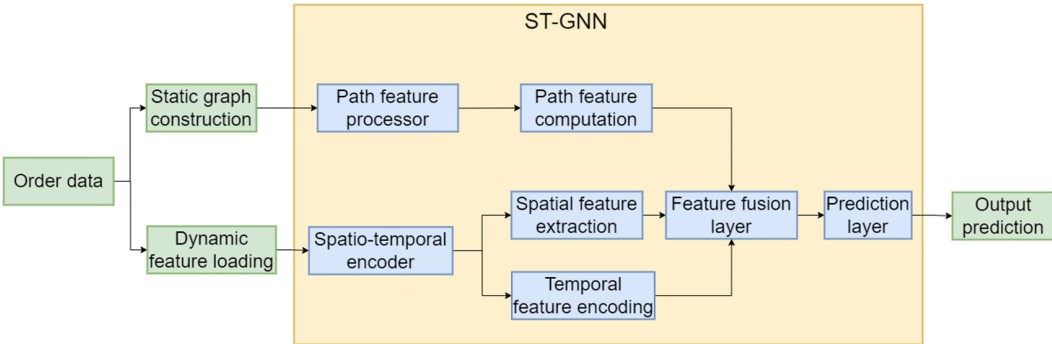

**Fig 1. Frame diagram of the ST-GNN model.**

## Path optimization algorithm

In order to realize the multi-objective dynamic path planning of double orders, the improved Dijkstra algorithm is used. By dynamically fusing the spatio-temporal feature embedding vector generated by the ST-GNN model and the road network topology information, the objectives of carbon minimization and passenger experience (detour rate) were co-optimized under the premise of meeting the time window constraints and vehicle capacity constraints.

The algorithm uses a dynamic weight update mechanism enhanced by ST-GNN embeddings, combines real-time traffic state and static road network attributes to construct an edge weight function, and captures spatio-temporal dependencies through Multi-head Graph Attention Network (GAT) and Transformer encoder to realize accurate perception of dynamic traffic state in complex urban scenes. The enhanced edge weight calculation formula is as follows:

$$W^*(e_{ij}, t) = \alpha \cdot D_{ij} + \beta \cdot T_{ij}(t) + \gamma \cdot \text{MLP}(z_i(t) \oplus z_j(t) \oplus a_{ij}) \tag{1}$$

where: $W^*(e_{ij}, t)$: the enhanced weight of edge $e_{ij}$ at time t (passage cost);

$D_{ij}$: static distance from node i to j;

$T_{ij}(t)$: the dynamic travel time from node i to j at time t;

$z_i(t), z_j(t)$: spatio-temporal embeddings from ST-GNN output (128-dimensional);

$a_{ij}$: static edge attributes (road grade, number of roads, etc.);

$\oplus$: concatenation operation;

MLP: multi-layer perceptron with hidden layers [144, 64, 1];

$\alpha, \beta, \gamma$: learnable parameters with $\alpha + \beta + \gamma = 1$ (empirically set to 0.4, 0.4, 0.2).

Learnable Parameters:ST-GNN parameters: GAT layers (input_dim = 5→64, 64→64), LSTM (input_size = 4→64), Time embedding (1440→16);MLP parameters: Linear layers (144→64→1) with ReLU activation and dropout = 0.5;Balance weights: $\alpha, \beta, \gamma$ optimized during training

Normalization Procedures:Node features: StandardScaler for area, POI densities (Catering, accommodation services, corporate, transportation facilities);Edge weights: Min-max normalization to [0,1] range;Dynamic features: Exponential smoothing with decay factor 0.9 for temporal stability

The core of the algorithm is to balance the multi-objective conflict through the Pareto front strategy, and combine the local priority search with the global optimal path solution, so as to improve the global optimization ability of path planning while ensuring computational efficiency.

Online Inference Algorithm:

Input: Current time $t$, order pair $(o_k, o_l)$, ST-GNN model $M$.

Step 1:Get time slot $ts \leftarrow \lfloor (t - t_{\text{midnight}})/30 \rfloor$.

Step 2:Load dynamic features: $N_f \leftarrow$ node_features[$ts$], $E_f \leftarrow$ edge_features[$ts$].

Step 3:Generate ST-GNN embeddings: $Z \leftarrow M(N_f, E_f, ts)$.

Step 4:Update edge weights using Equation (1).

Step 5:Generate candidate paths: $P_{\text{candidates}} \leftarrow \{\text{Sequence A, Sequence B}\}$.

Step 6: For each path $p \in P_{\text{candidates}}$: Compute total cost: $C(p) \leftarrow \sum_{e_{ij} \in p} W^*(e_{ij}, t)$; Check constraints: time windows, detour ratio $\leq 0.4$.

Step 7: Select optimal path: $P^* \leftarrow \arg\min_p C(p)$.

Step 8: Compute matching score: $s \leftarrow \text{sigmoid}(MLP(Z_k \oplus Z_l))$.

Output: Optimal path $P^*$, matching score $s$.

Time Complexity Analysis:ST-GNN inference: $O(|V| + |E|)$ per time slot; Edge weight update: $O(|E|)$; Path planning: $O((|V| + |E|) \log |V|)$ using priority queue;Total complexity: $O((|V| + |E|) \log |V|)$ per order pair.

## Objective function and constraints

This paper studies the efficient matching and dynamic path planning of orders through spatio-temporal awareness graph neural network to minimize carbon emissions under the premise of meeting time window constraints and vehicle capacity constraints.Objective function:

$$\min \left[ \beta d(P_k) + \beta d(P_l) - \beta d(P_{kl}) \right] \qquad (2)$$

$P_{kl}$: represents the joint path of order pair (k,l); $d(P_k)$,$d(P_l)$ : are the independent path distances of orders k and l, respectively; $d(P_{kl})$ : is the distance between order k and l carpooling path;$\beta$: is the carbon emission factor per unit distance (kg $CO_2$/km).

Constraints:

$$\text{idx}(P_i^{\text{pickup}}) < \text{idx}(P_i^{\text{dropoff}}), \quad \forall o_i \subset O \qquad (3)$$

$$P_i^{\text{pickup}} > T_i^{\text{max}}, \quad \forall o_i \subset O \qquad (4)$$

idx(v) : the position index of node v in the path sequence; $P_i^{\text{pickup}}$: the pick-up section of order oi; $P_i^{\text{dropoff}}$: Disembarkation node of order oi; $T_i^{\text{max}}$: The latest allowable loading time of order oi.

Objective function (2) aims to find the optimal path by minimizing the detour rate and minimizing the carbon emissions. Constraint: Equation 3 indicates that the allocator node of each order must appear after the allocator node. Equation 4 indicates that the actual pickup time of a passenger must not exceed his promised latest waiting time.

## Algorithm flow

The double order pickup path optimization algorithm based on ST-GNN proposed in this study combines the spatio-temporal perception ability of graph neural network with the dynamic path planning mechanism, and the overall process is as follows.

Step 1: One-Pass clustering is performed on the pick-up point and drop-off point of the online car-hailing order, and the center point of the hot area is generated as the graph node.

Step 2: The center of each region is used as the graph node $v \in V$. The road network connection is defined as the graph edge $e_{ij} \in E$, and the edge weight is initialized as the historical average travel time. Node features combine static attributes (area, POI density) and dynamic attributes (real-time order quantity), and edge features include road section length, average travel time, etc.

Step 3: The edge weights (travel time) and node characteristics (order quantity) of the graph are dynamically updated according to the order timestamp, and the time-varying graph Gt = (V,E,W t) is constructed.

Step 4: Load order data from CSV file, build spatial index through BallTree, convert geographical coordinates into graph node index, and at the same time: divide the day into multiple time slots (one time slice every 30 minutes).

Step 5: The historical matching data is used to generate positive samples (real matching order pairs and their generated joint paths), and the non-matching order pairs are generated by random sampling and path validity filtering (such as time window conflicts) to generate negative samples.

Historical Optimal Path Definition: The historical optimal path is defined as the shortest travel-time path under historical median speeds for the same OD pair during the same time period, incorporating real traffic conditions including congestion factors. Specifically:

1. For each origin-destination pair $(s_i, d_i)$ in time slot $t$, we compute the median travel time from historical trajectory data.

2. The path with minimum median travel time is selected as the historical optimal path $P_i^{opt}$.

3. This path incorporates actual traffic signals, congestion patterns, and road conditions observed in historical data.

Step 6: Multi-head GAT (Graph Attention Network) is used to aggregate neighborhood information, calculate attention coefficients, and fuse static and dynamic node/edge features. At the same time, TemporalConv (temporal Convolutional layer) is used to extract the time series pattern of node features (such as the morning and evening peak law), and the dynamic changes are modeled by combining historical time slice data. The gated mechanism dynamically balances the spatio-temporal features, generates 128-dimensional spatio-temporal embedding vectors, and retains the original feature information.

Step 7: The matching probability of the order pair is calculated based on the spatio-temporal embedding vector, and compared with the actual matching label to optimize the matching ability of the model.

Loss Function Construction: The training objective combines matching classification and path quality regression:

$$\mathcal{L} = \lambda_1 \cdot \mathcal{L}_{match} + \lambda_2 \cdot \mathcal{L}_{detour} \tag{5}$$

where: $\mathcal{L}_{match}$: Binary cross-entropy loss for order pairing probability prediction;

$\mathcal{L}_{detour}$: Mean squared error between predicted detour rate and actual detour rate relative to historical optimal path;

$\lambda_1, \lambda_2$: Balance weights

Step 8: For the order pair (ok,ol), two kinds of joint path candidates are generated: sequence A: sk→sl→dk→dl sequence B: sk→sl→dl→dk. The connectivity of each subpath (sk→sl, sl→dk, dk→dl, etc.) is verified in the road network graph by Dijkstra algorithm and merged into a complete path.

Step 9: The matching probability of the order pair is calculated based on the spatio-temporal embedding vector, and compared with the actual matching label to optimize the matching ability of the model. At the same time, for the generated joint path, the deviation between the predicted detour rate and the actual detour rate (the historical optimal path) was calculated to minimize the path planning error.

## Experiment and result analysis

### Data and parameter settings

This data selects the Didi order data and vehicle trajectory data in Chengdu City: it covers the order and order trajectory records from November 1, 2016 to November 30, 2016. The data volume of the order data is about 51MB after preprocessing, and a total of 370,000 data. The dataset includes attributes such as order ID, start billing time, end billing time, longitude of pick-up location, latitude of pick-up location, longitude of drop-off location, and latitude of drop-off location. After preprocessing, the amount of vehicle trajectory data is about 8.5GB, with a total of 74.52 million data. This dataset includes attributes such as driver ID, order ID, GPS time, longitude of the trajectory, latitude of the trajectory point, and so on.

Model parameters: STGNN hidden layer Dimension: hidden_dim = 64. Long time window: time_window = 30, chosen to balance the capture of short-term traffic fluctuations with the need for sufficient data density for stable modeling. Number of attention heads: GATConv is set to 2 heads. Dropout rate: dropout = 0.1.

Path planning parameters: Maximum detour rate threshold: max_detour = 0.4. Unit carbon emission factor: $\beta$ = 0.2 kg/km. The carbon emission factor $\beta$ is set to a constant value of 0.2 kg $CO_2$/km. While real-world emissions are influenced by speed, congestion, and vehicle type, the use of a fixed factor is a common simplification in macroscopic ride-pooling studies. It serves as a effective proxy for minimizing total travel distance, which is the primary driver of fuel consumption and emissions at the system level.The spatial clustering radius of 500 meters was selected based on typical urban walking accessibility, ensuring geographical proximity for feasible ride-pooling while maintaining appropriate spatial granularity.

## Data split and leakage prevention strategy

To ensure no information leakage and maintain temporal integrity, we implemented a strict chronological data splitting strategy:

Temporal Splits:Training set: November 1-21, 2016 (21 days, 70% of total orders);Validation set: November 22-27, 2016 (6 days, 20% of total orders);Test set: November 28-30, 2016 (3 days, 10% of total orders).

Spatial Partitioning:The Voronoi polygon spatial partitioning was generated exclusively using the training set data (November 1-21) to prevent any test set information from influencing the spatial structure. The same spatial partitioning was then applied to validation and test sets without modification.

Leakage Prevention Measures:

1. Strict Temporal Ordering: We employed a chronological data splitting where the model is trained on earlier data and evaluated on later data, ensuring no future information is used during training.

2. Dynamic Feature Construction: For each time slice, dynamic features (e.g., real-time order volume, travel time) are computed using the actual observed data within that specific time slice. This approach uses contemporaneous rather than predicted values for dynamic features.

3. Model Training Protocol: The ST-GNN model was trained exclusively on the training set (Nov 1-21), with hyperparameter tuning performed on the validation set (Nov 22-27). The final evaluation was conducted on the untouched test set (Nov 28-30, 5,460 trips).

4. Feature Scope Clarification: It is important to note that the dynamic features (real-time order volume, travel time) used in this study are based on actual observations within each time slice. The prediction of these dynamic features from historical data is considered as a direction for future research.

## Definition of evaluation indicators

In order to comprehensively evaluate the effectiveness of the proposed ST-GNN-based double-order ride-pooling method, this study selected the following three core evaluation indicators: matching success rate, average carbon emission reduction per order, and average detour rate. These indicators measure the performance of the algorithm from three dimensions: order matching efficiency, environmental protection benefit and passenger experience.

1. Matching success rate (MSR): the proportion of the number of successfully matched order pairs in the total order number reflects the efficiency of the algorithm in order matching. The formula is as follows:

$$\text{MSR} = \frac{M}{N} \times 100\% \tag{6}$$

2. Average carbon emission reduction per order (ACERO): The carbon emission reduction per order through the ride-pooling path reflects the contribution of the algorithm in terms of environmental benefits, and is calculated as follows:

$$\text{ACERO} = \frac{1}{M} \sum_{k,l \in P}^{M} [\beta d(P_k) + \beta d(P_l) - \beta d(P_{kl})] \tag{7}$$

3. Average detour rate (ADR): The average detour ratio of all carpooling orders, measuring the additional driving distance that passengers travel due to carpooling. The calculation formula is as follows:

Let $P$ be the number of successfully matched order pairs, and $2P$ be the total number of orders (since each pair contains two orders).

For each order $i$ in a matched pair, the detour rate is calculated as:

$$r_i = \frac{D_i^{\text{actual}} - D_i^{\text{base}}}{D_i^{\text{base}}}$$

The average detour rate across all $2P$ orders is:

$$\text{ADR} = \frac{\sum_{i=1}^{2P} r_i}{2P}$$

Where: $D_i^{\text{base}}$: The independent path distance of order $i$; $D_i^{\text{actual}}$: The actual carpooling route distance of order $i$; $r_i$: The detour rate of order $i$; $P$: Number of successfully matched order pairs; $2P$: Total number of orders in all matched pairs.

Worked Example:Consider a scenario with $P = 3$ successfully matched order pairs (thus $2P = 6$ total orders). The detour rates for each order are: - Order Pair 1: $r_1 = 0.15$, $r_2 = 0.10$ - Order Pair 2: $r_3 = 0.20$, $r_4 = 0.12$ - Order Pair 3: $r_5 = 0.08$, $r_6 = 0.18$

The average detour rate is calculated as:

$$\text{ADR} = \frac{0.15 + 0.10 + 0.20 + 0.12 + 0.08 + 0.18}{6} = \frac{0.83}{6} = 0.1383$$

This means passengers experience an average detour of approximately 13.83% due to ride-pooling.

## Experimental analysis

In order to verify the effectiveness of the proposed two-passenger ride-pooling method based on ST-GNN and path optimization, experiments are carried out on the order data and order trajectory data of Didi in Chengdu, and the performance of different algorithms is compared.

1) Data mining and regional division based on urban road network.First, The order data of Didi in Chengdu in November 2016 was preprocessed, integrate the data set containing order information and driver information, and perform deduplication operations to ensure data quality. Then, we perform data join and grouping operations based on "driver ID" to associate the order with its execution driver. The core identification logic lies in the time overlap detection: the order set belonging to the same driver is sorted by the "start billing time", and whether there is an overlap between the order time interval (start billing time to end billing time). According to the chronological order, R2 mode can be formed if there are adjacent orders, and orders are merged to determine whether R(x > 2) exists one by one, so as to calculate the distribution of the number of drivers under different x values (i.e. Rx mode), as shown in Fig 2.

From the figure, it is found that the number of drivers with two-passenger ride-pooling orders ($R_2$) occupies an absolute advantage. By counting the number of associated orders x and generating a histogram, the results show that x = 2 is a significant peak, and the proportion of double order scenario ($R_2$) is as high as 92.8%. This finding indicates that in practice, double pooling is the most dominant form of ride-pooling, which achieves a good balance between resource utilization efficiency and algorithm complexity. Therefore, this paper focuses on the double order scenario ($R_2$) with x = 2, in order to optimize the matching and path planning strategy.

After determining the research focus on double booking orders ($R_2$), we further mine this part of the data in depth, aiming to identify its inherent pick-up and drop timing patterns. For all drivers in $R_2$ mode, we extract the two associated orders and sort them by their "start billing time." Subsequently, by comparing the "end billing time" order of these two orders, we identify two core pick-up and drop-off timing patterns: $R_2$1 pattern (first up, first down) and $R_2$2 pattern (first up, then down), as shown in Fig 3.

 

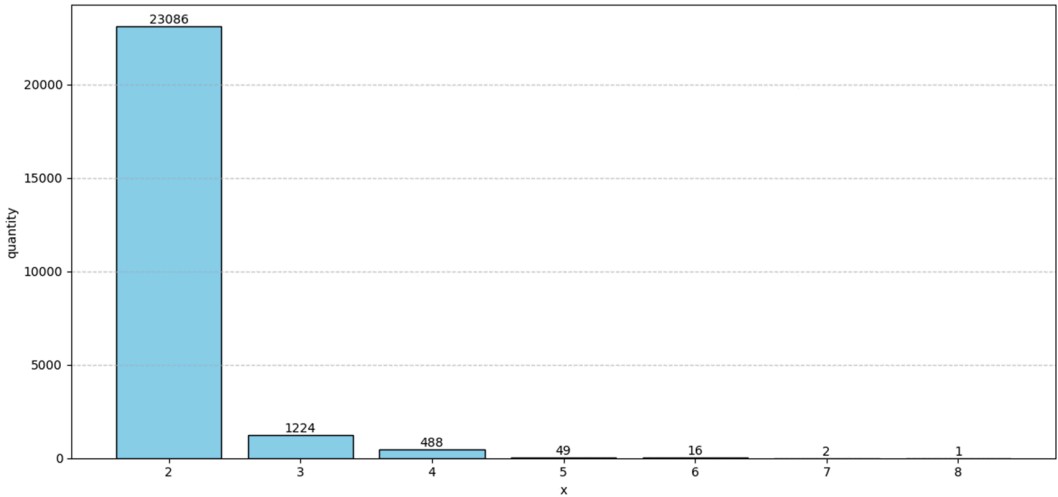

**Fig 2**. **Rx distribution diagram.**

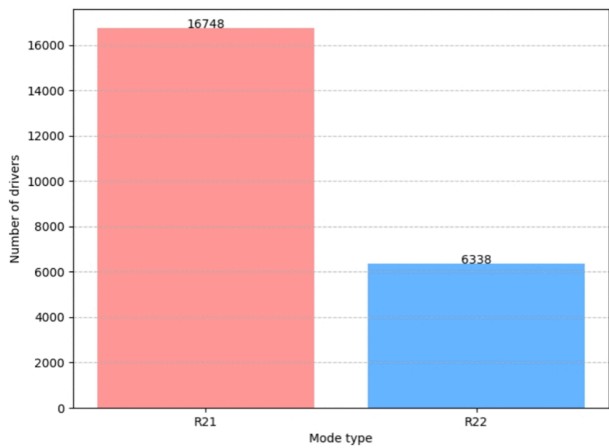

**Fig 3**. **Distribution of R$_{xy}$.**

In order to understand the spatial distribution characteristics of two-passenger ride-pooling ($R_2$) more deeply, and provide a basis for constructing a regionalized spatio-temporal graph model. The improved One-Pass clustering algorithm is used to perform density clustering on the pick-up points of double booking orders (parameter Settings: spatial neighborhood radius r = 500 meters, minimum number of clustering samples MinPts = 50), and high-frequency travel demand areas are identified. In the experiment, a total of 94 core cluster centers were extracted (Fig 4), and at the same time, the Amap POI data were integrated to calculate the four major types of facility density in each region (Fig 5), in which the regions in the poi dataset showed significant high density characteristics (light yellow area).

The center point of the cluster is used as the seed to generate the Voronoi polygon (Fig 6), and the region boundary is divided by the vertical bisector adjacent to the center point to form the spatial topology of the urban demand distribution. The area and shape of each region reflect regional heterogeneity, the color intensity represents the number of boarding points, and the red dot is the cluster center. And from the area and shape of the Voronoi polygon reflects the heterogeneity of the region. For example, regions with high POI density are divided into smaller Voronoi polygons, while suburbs

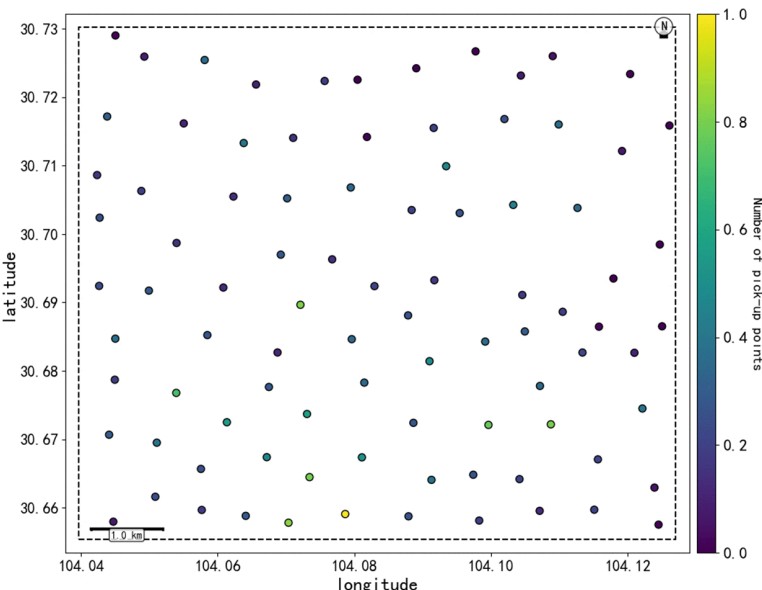

**Fig 4**. **Clustering diagram of boarding points.**

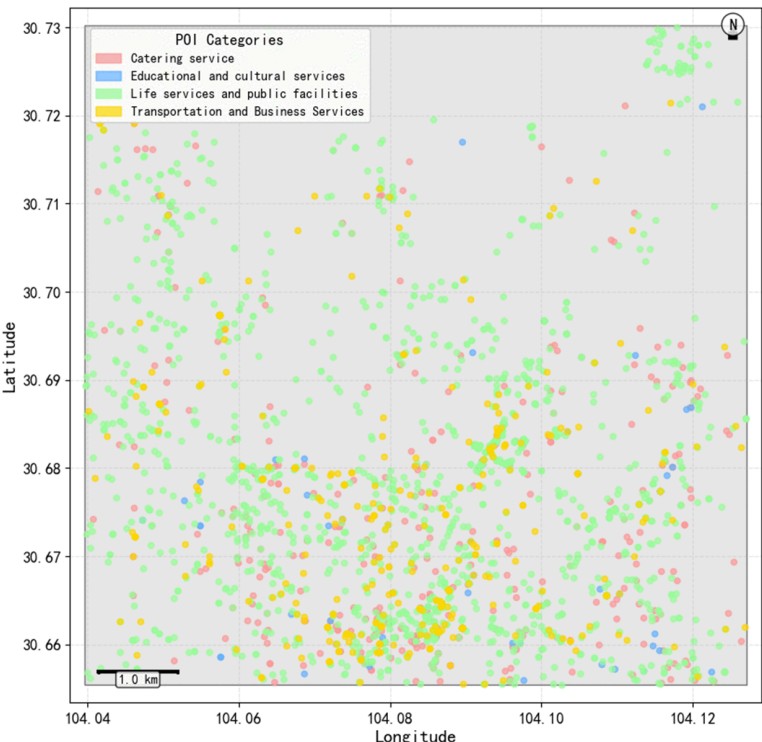

**Fig 5**. **Distribution of poi.**

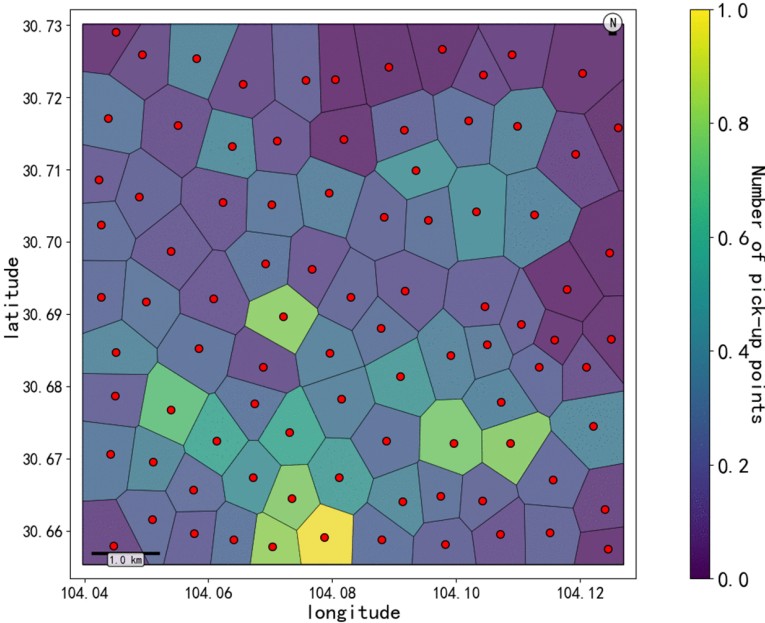

**Fig 6**. Heat map of Voronoi polygon based on density points on board.

with low POI density may form larger polygons, which can be seen from the denser Voronoi polygons in the lower part of Fig 6.

After the urban region division is completed, a spatio-temporal graph model integrating static attributes and dynamic characteristics is further constructed. The center of each Voronoi polygon is taken as the graph node, and the connectivity between regions is defined as the graph edge, as shown in Fig 7. By fusing static attributes (such as POI density, number of roads between edges, road level, etc.) and dynamic attributes (such as real-time order quantity, travel time), the graph structure with spatio-temporal awareness is constructed. Taking 30 minutes as a time slice, the dynamic update mechanism periodically adjusted the edge weights and node characteristics according to real-time vehicle trajectory data and order distribution characteristics to ensure that the model could accurately reflect the current traffic state.

2) Order matching and path planning.In order to verify the performance of the constructed ST-GNN model in actual order matching and route generation, 5460 ride-hailing data from November 28 to 30, 2016 in Chengdu were selected for model testing. The experiments covered different periods (morning peak, flat peak, evening peak) and regional types (commercial district, residential district, mixed district).

Firstly, the BallTree spatial index is established for all the order data, and the BallTree spatial index and time window constraint are used to screen the candidate order pairs, and the improved Dijkstra algorithm is used to generate the independent path node sequence according to the OD information of the order. Then, the spatio-temporal embedding vector of the order path is extracted through the trained ST-GNN model, and the matching score of the order pair is generated by combining the time slice of the order, the path node sequence and the dynamic graph features. And the order pairs with higher matching scores are screened out as potential ride-pooling combinations. Then, the improved Dijkstra path planning algorithm was called to search for the optimal path in the urban traffic graph divided by Voronoi polygon. Then, two candidate path orders (first receive first send, first receive then send) are generated, and the complete path is generated by sub-path extraction and path splicing. After the path is generated, the quality of the path is evaluated, the total distance traveled, carbon emissions, detour rate and other indicators are calculated, and the path validity is judged by combining the time window constraint. Finally, the optimal ride-pooling path scheme satisfying the constraints was output.

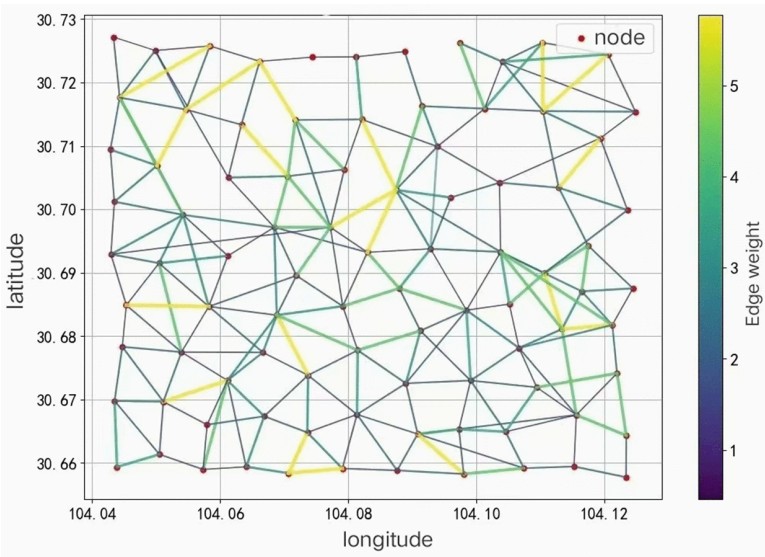

**Fig 7. Spatio-temporal diagram of urban road network based on Voronoi polygon.**

The matching scheme and planning path generated by 5460 ride-pooling network travel data were evaluated. To ensure the statistical reliability of the results, each experiment was repeated for 10 independent trials with different random seeds. As shown in Table 1 below:

Some orders in the noon period of 29th (13:30-14:00) are selected for case illustration. The order information is shown in Table 2:

First, the order trajectory is mapped onto the spatio-temporal graph, as shown in Fig 8. Then, the time window constraint and ST-GNN model were used to form the order pair according to the path similarity, and the carpool path of the order was generated (as shown in Fig 9). Based on the model's prediction, a joint path was generated for the order pair, which resulted in a reduction of carbon emissions and the reduction of detour rate under the premise of meeting the time window constraint.

**Table 1. Performance comparison of different path planning algorithms combined with ST-GNN.**

| Algorithm Combination | Matching success rate (%) | verage carbon emission reduction per order ($kgCO_2$) | Average detour rate |
|---|---|---|---|
| ST-GNN + Floyd | $82.5 \pm 0.8$ | $0.28 \pm 0.013$ | $0.1580 \pm 0.0069$ |
| ST-GNN + A* | $84.8 \pm 0.7$ | $0.29 \pm 0.017$ | $0.1430 \pm 0.0058$ |
| ST-GNN + Modified Dijkstra | $86.4 \pm 0.5$ | $0.33 \pm 0.015$ | $0.1195 \pm 0.0035$ |

**Table 2. Order information.**

| Order | Start and end node | Order start time | Allow the latest fetch time | Sequence of travel path nodes |
|---|---|---|---|---|
| 1 | (59,34) | 13:30:30 | 13:35:30 | 59,61,43,40,37,34 |
| 2 | (58,35) | 13:42:52 | 13:47:52 | 58,61,43,40,37,35 |
| 3 | (73,37) | 13:30:27 | 13:35:27 | 73,64,56,48,37 |
| 4 | (67,34) | 13:35:45 | 13:40:45 | 67,63,49,37,34 |
| 5 | (67,80) | 13:37:34 | 13:42:34 | 67,64,73,80 |
| 6 | (17,49) | 13:42:28 | 13:47:28 | 17,28,38,40,49 |
| 7 | (35,26) | 13:46:02 | 13:51:02 | 35,34,36,38,26 |
| 8 | (17,32) | 13:30:03 | 13:35:03 | 17,28,32 |

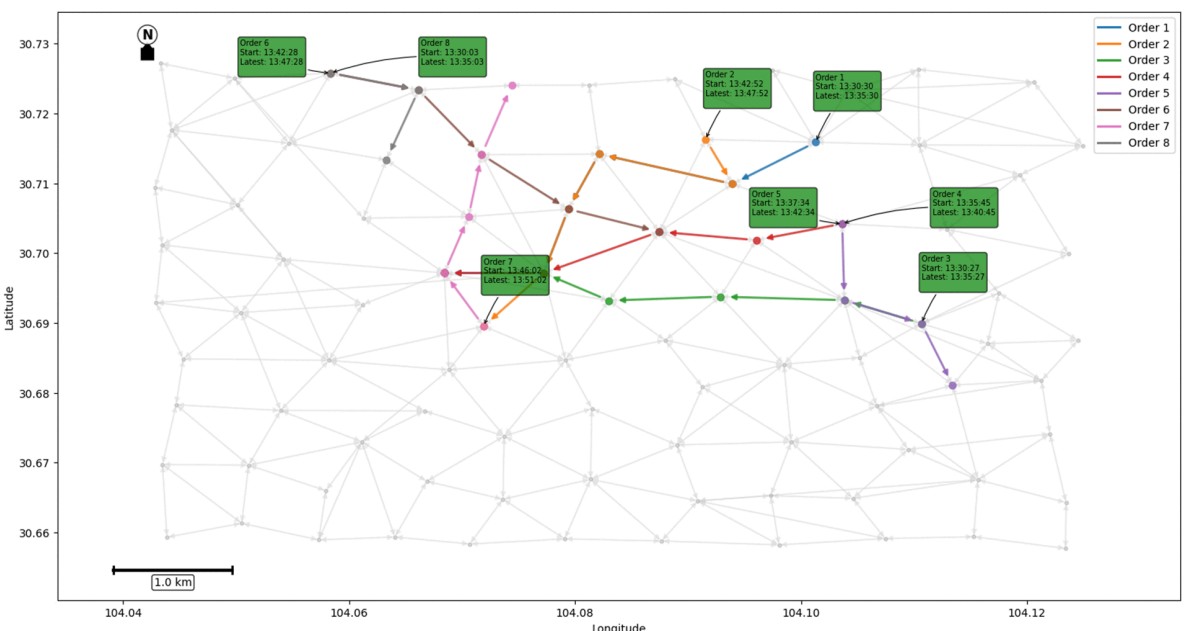

**Fig 8. Independent paths of orders.**

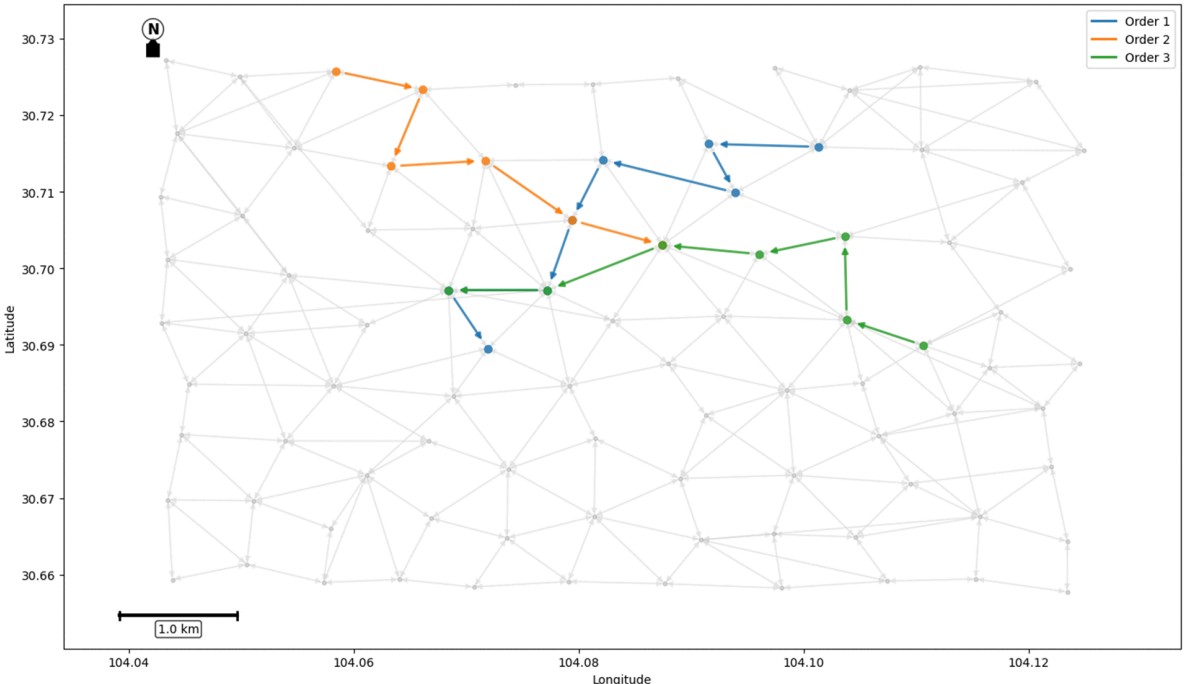

**Fig 9. The carpool path of the order.**

**Table 3. Matching results.**

| Matched orders | Path sequence | Carbon emission (kg $CO_2$) | Detour rate 1 | Detour rate 2 |
|---|---|---|---|---|
| Order 1, Order 2 | 59,58,61,43,40,37,34,35 | 1.86 | 0.128 | 0.156 |
| Order 3, Order 4 | 73,64,67,63,49,37,34 | 1.34 | 0.253 | 0.103 |
| Order 6, Order 8 | 17,28,32,38,40,49 | 1.08 | 0.143 | 0 |

By comparing Figs 8 and 9, it is obvious that the ride-pooling path reduces the total travel distance as a whole and effectively avoids unnecessary detaches.

The experimental results are shown in Table 3. The carpooling scheme successfully matched three pairs of orders, and on average, each order reduced carbon emissions by 0.23 kg of $CO_2$.

## Conclusion

This study proposes an optimization for two-passenger ride-pooling orders method based on ST-GNN (spatio-temporal perception graph neural network) and path optimization, aiming to solve the problem of dynamic ride-sharing matching and path planning in urban traffic. By constructing a spatio-temporal graph model that integrates the Voronoi polygon regional topology and dynamic order characteristics, and combining the multi-dimensional attention mechanism, the coupling modeling of static road network attributes and dynamic traffic states is realized, which effectively improves the success rate of ride-pooling order matching and the global optimization ability of path planning.

The experimental results show that the proposed method performs well on the actual order data of Chengdu. Based on the combination of ST-GNN and the improved Dijkstra algorithm, the matching success rate of 5460 ride-pooling network travel data from November 28 to 30, 2016 in Chengdu reaches 86.6%, the average carbon emission reduction is 0.34 kg $CO_2$ per order, and the average detour rate is only 0.1202. This result supports the adaptability and practicability of the model in complex urban traffic scenes. At the same time, the case study further shows that the proposed method can effectively support the dynamic ride-pooling of short-distance orders under the dense road network in the commercial area, which can significantly reduce the driving distance and carbon emissions, while ensuring the time window constraints of passengers.

### Limitations and future work

This work has limitations that point to clear future directions. The use of actual observed data for dynamic features (e.g., travel time) demonstrates ideal performance but requires prediction for practical deployment. Furthermore, the fixed carbon emission factor, while effective for minimizing distance, overlooks variations due to speed and congestion. The development of predictive models for dynamic features and the integration of a speed-dependent emission factor $\beta(v)$ are identified as immediate priorities for future research.

## Author contributions

**Conceptualization:** Xue Xing, Yuqi Peng.

**Data curation:** Yuqi Peng.

**Formal analysis:** Yuqi Peng.

**Funding acquisition:** Xue Xing.

**Investigation:** Yuqi Peng.

**Methodology:** Yuqi Peng.

**Project administration:** Xue Xing.

**Resources:** Xue Xing.

**Software:** Yuqi Peng.

**Supervision:** Xue Xing.

**Validation:** Yuqi Peng, Le Wan, Fahui Luo.

**Visualization:** Yuqi Peng, Le Wan.

**Writing – original draft:** Yuqi Peng.

**Writing – review & editing:** Yuqi Peng.

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
