## [Decision Letter · Decision Letter 0]

23 Sep 2025

PONE-D-25-48123Double-Pooling Optimization of Ride-hailing Orders Based on ST-GNN and Path OptimizationPLOS ONE

Dear Dr. Xing,

Thank you for submitting your manuscript to PLOS ONE. After careful consideration, we feel that it has merit but does not fully meet PLOS ONE’s publication criteria as it currently stands. Therefore, we invite you to submit a revised version of the manuscript that addresses the points raised during the review process.

We look forward to receiving your revised manuscript.

Kind regards,

Guangyin Jin

Academic Editor

PLOS ONE

Journal Requirements:

3. We note you have included a table to which you do not refer in the text of your manuscript. Please ensure that you refer to Tables 1, 2 and 3 in your text; if accepted, production will need this reference to link the reader to the Table.

 “This work was supported by the Industrialization Cultivation Project of the Jilin Provincial Department of Education [Grant Number JJKH20230306CY].”        

5. Please note that your Data Availability Statement is currently missing direct link to access each database. If your manuscript is accepted for publication, you will be asked to provide these details on a very short timeline. We therefore suggest that you provide this information now, though we will not hold up the peer review process if you are unable.

Reviewers' comments:

Reviewer's Responses to Questions

**Comments to the Author**

1. Is the manuscript technically sound, and do the data support the conclusions?

Reviewer #1: Yes

Reviewer #2: Partly

2. Has the statistical analysis been performed appropriately and rigorously?

Reviewer #1: Yes

Reviewer #2: No

3. Have the authors made all data underlying the findings in their manuscript fully available?

Reviewer #1: Yes

Reviewer #2: Yes

4. Is the manuscript presented in an intelligible fashion and written in standard English?

Reviewer #1: Yes

Reviewer #2: Yes

5. Review Comments to the Author

Reviewer #1: 1.The core of the paper lies in how the ST-GNN embedding modifies edge weights before entering the “modified Dijkstra” module, but the manuscript lacks an explicit mathematical definition and pseudocode.

Please provide an edge-weight mapping of the form

we∗(t)=α westatic+γ f(zt,ae,…)w_e^{*}(t)=\alpha\,w^{\text{static}}_e+\gamma\,f(\mathbf{z}_t,\mathbf{a}_e,\ldots)we∗(t)=αwestatic+γf(zt,ae,…),

list all learnable parameters, describe normalization/stabilization procedures, and include complete online inference pseudocode with time complexity O(⋅)O(\cdot)O(⋅).

2.The training objective is only described vaguely (“use spatio-temporal embeddings to estimate pairing probability; minimize the deviation between predicted detour rate and the historical optimal path”), but the “historical optimal path” is not defined.

Please formalize how labels are generated and how the loss is constructed: Is the historical optimal path a static shortest path, a shortest travel-time path under historical median speeds, and does it incorporate signals/congestion factors?

3.In Table 1 you compare “ST-GNN + Floyd/A*/modified Dijkstra.” If these methods use different edge weights (i.e., not the same dynamic cost definition), the comparison is not fair.

Please standardize and disclose the cost function, tuning protocol, and stopping criteria across all solvers.

4. The manuscript states that features are built from the entire November 2016 data, while testing uses Nov 28–30 (5,460 trips). If ST-GNN training also uses dynamics from those test days, you must demonstrate there is no information leakage (e.g., a strict left-closed/right-open rolling window; no use of future traffic states).

Please specify train/validation/test temporal splits, spatial partitioning, and any cross-validation strategy.

5. The unit carbon factor is fixed at β=0.2 kg/km\beta=0.2\ \text{kg/km}β=0.2 kg/km, which ignores the effect of speed/idling/congestion on CO₂. I suggest:

a) Cite authoritative emission factors or define a speed-dependent β(v)\beta(v)β(v), and provide a sensitivity analysis;

b) In the case study, demonstrate the robustness of the chosen optimal path when β\betaβ varies.

6. In the ADR formula the denominator M×2M\times 2M×2 implicitly assumes “two orders per pair,” yet MMM is used to indicate both “number of pairs” and “number of samples,” which is confusing.

Please unify the notation as PPP pairs and 2P2P2P orders, and provide a worked example.

7. Replace “Tyson polygon” with “Thiessen (Voronoi) polygons”, and on first use provide the mathematical definition and construction details (adjacency rule, prohibition of edge crossing).

8. Terminology unification: choose a single term among carpooling / ride-pooling / ride-sharing; in the title, standardize “Double-Pooling/dual-pool/two-order pooling” to “two-passenger ride-pooling.”

9. Units and formatting: use kg CO2_22 (with a space before the unit). In Table 1, use the column header “Average carbon emission reduction per order (kg CO2_22)”.

10. Maps/route figures: add a scale bar, north arrow, and basemap attribution; for Figs. 8/9, annotate time windows and constraint feasibility.

11. Tone and references: remove promotional language and replace with verifiable statements; ensure journal titles and publication years are formatted and reported consistently.

Reviewer #2: This paper proposes a dual-optimization approach for ride-sharing order matching that combines a spatio-temporal graph neural network (ST-GNN) with an improved Dijkstra algorithm. The authors use Tyson polygons for spatial division of Chengdu's urban area and apply multi-head graph attention networks with Transformer encoders to capture spatio-temporal dependencies. Testing on Didi order data from November 2016, the method achieves an 86.6% matching success rate with 0.34kg CO2 reduction per order.

While the integration of graph neural networks with ride-sharing optimization presents an interesting direction, several fundamental issues need addressing before this work can make a solid contribution to the field.

1. The ST-GNN architecture description lacks mathematical formulation and technical depth. Section 3's "The ST-GNN model framework" provides only high-level descriptions without explaining the specific layer configurations, loss functions, or training procedures. For instance, how exactly does the "gated mechanism" in Step 6 balance spatial and temporal features? What objective function guides the model training? The paper mentions generating a "128-dimensional spatio-temporal embedding vector" but never explains how this dimensionality was chosen or validated. This vagueness makes the work difficult to reproduce or properly evaluate.

2. The comparison is limited to combining ST-GNN with only three path-finding algorithms (Floyd, A*, modified Dijkstra), which doesn't demonstrate the ST-GNN's actual contribution versus traditional ride-sharing methods. Where are comparisons with established ride-sharing algorithms like those by Alonso-Mora et al. (2017) on dynamic vehicle routing, or Ma et al. (2013) on T-Share? The authors cite several ride-sharing papers in the introduction but never compare against them experimentally. Additionally, testing on only 5,460 orders from three days is insufficient to validate a method intended for large-scale deployment.

3. Table 1's results lack any statistical significance testing, confidence intervals, or standard deviations across multiple runs. Did the authors perform cross-validation? How sensitive are results to different data splits or time periods? The claimed improvements (86.6% vs 85.1% matching rate) could easily fall within statistical noise. Without proper statistical analysis, readers cannot assess whether the reported improvements are meaningful or merely random variations.

4. The paper provides no discussion of computational complexity or scalability. What are the time and space complexities of the proposed approach? How does runtime scale with the number of orders, graph nodes, or time windows? For a real-time ride-sharing system, these considerations are crucial. The authors mention "ensuring computational efficiency" but provide no actual runtime measurements or complexity analysis.

5. The abstract is overly dense with technical details while the introduction fails to clearly position the work within existing literature. Section 2 jumps between problem definition and methodology without clear transitions. The conclusion merely restates results without discussing limitations or future work. Consider restructuring to follow a clearer narrative: problem → existing solutions → gaps → proposed method → validation.

6. Figures 4-6 showing clustering and Tyson polygons lack scale bars and proper legends. Figure 1's system architecture diagram uses vague terms like "Feature fusion layer" without explaining what features are being fused or how. Figure 8 and 9 comparing independent versus pooled paths need clearer visualization - perhaps showing the actual routes on a map rather than abstract node connections.

7. Why use Tyson polygons over grid-based or administrative boundary divisions? The authors claim Tyson polygons better represent "real demand distribution" but provide no comparative analysis. Similarly, the choice of 30-minute time windows, 500-meter clustering radius, and other parameters appears arbitrary without sensitivity analysis or justification.

8. The carbon reduction calculation uses a simple linear factor (β = 0.2kg/km) without considering vehicle types, traffic conditions, or actual emission patterns. Real-world emissions vary significantly based on speed, acceleration patterns, and vehicle occupancy. This oversimplification undermines the environmental claims.

9. The manuscript contains numerous grammatical errors and awkward phrases. Examples include "Through spatio-temporal perception modeling" (missing subject), "AmAP POI data" (likely meant "Amap"), and inconsistent terminology (sometimes "carpool," sometimes "carpooling," sometimes "ridesharing"). These issues detract from the technical content and suggest insufficient proofreading.

10. The paper lacks crucial details for reproducibility. What framework was used for the GNN implementation? How were hyperparameters selected? What hardware was used for experiments? The GitHub repository or code availability is not mentioned, making it impossible to verify results or build upon this work.

6. PLOS authors have the option to publish the peer review history of their article (what does this mean?). If published, this will include your full peer review and any attached files.

Reviewer #1: No

Reviewer #2: No

---

## [Author Response · Author response to Decision Letter 1]

6 Nov 2025

We thank the editors and reviewers for their valuable comments and suggestions. We have thoroughly revised the manuscript accordingly and provided a detailed point-by-point response to all comments in the uploaded file "Response to Reviewers.pdf". All changes in the manuscript are highlighted in the revised version for ease of review.Additionally, as requested, we have amended the Cover Letter to include the role of the funder statement: "The funders had no role in study design, data collection and analysis, decision to publish, or preparation of the manuscript."

---

## [Decision Letter · Decision Letter 1]

10 Nov 2025

Optimization of Two-Passenger Ride-Pooling Orders Based on ST-GNN and Path Optimization

PONE-D-25-48123R1

Dear Dr. Xing,

We’re pleased to inform you that your manuscript has been judged scientifically suitable for publication and will be formally accepted for publication once it meets all outstanding technical requirements.

Kind regards,

Guangyin Jin

Academic Editor

PLOS ONE

Additional Editor Comments (optional):

Reviewers' comments:

Reviewer's Responses to Questions

**Comments to the Author**

1. If the authors have adequately addressed your comments raised in a previous round of review and you feel that this manuscript is now acceptable for publication, you may indicate that here to bypass the “Comments to the Author” section, enter your conflict of interest statement in the “Confidential to Editor” section, and submit your "Accept" recommendation.

Reviewer #1: All comments have been addressed

Reviewer #2: All comments have been addressed

2. Is the manuscript technically sound, and do the data support the conclusions?

Reviewer #1: Yes

Reviewer #2: Yes

3. Has the statistical analysis been performed appropriately and rigorously?

Reviewer #1: (No Response)

Reviewer #2: Yes

4. Have the authors made all data underlying the findings in their manuscript fully available?

Reviewer #1: Yes

Reviewer #2: Yes

5. Is the manuscript presented in an intelligible fashion and written in standard English?

Reviewer #1: Yes

Reviewer #2: Yes

6. Review Comments to the Author

Reviewer #1: No other concerns.

Reviewer #2: This manuscript proposes a dual-optimization framework for two-passenger ride-pooling. The authors construct a demand-adaptive urban graph using Voronoi polygons and employ a spatio-temporal GNN (ST-GNN) to learn embeddings. These embeddings inform both order matching and a multi-objective path optimization algorithm. Validated on a real-world dataset, the method demonstrates strong performance in matching success, detour reduction, and carbon savings, offering a practical solution for shared mobility.

The authors have done a commendable job of addressing the previous round of feedback, and the manuscript is significantly stronger as a result. The additions clarifying the methodology, such as the explicit edge weight formulation (Eq. 1), the loss function (Eq. 5), and the data splitting strategy, have greatly improved the paper's rigor. I have just a couple of minor suggestions for final polishing.

1. I was pleased to see the addition of the "Limitations and Future Work" section, which properly addresses the fixed carbon emission factor. The authors are correct that this is a common simplification. As they move forward with this research, I would suggest they consider not just a speed-dependent factor, but also factors related to vehicle acceleration and idling, which are significant contributors to emissions in congested urban environments. This would be a strong next step for refining the environmental optimization aspect.

2. The enhanced description of the ST-GNN model framework is much clearer. I noted the use of Transformer encoder layers for temporal modeling. While the paper mentions this captures "long-range dependencies," a brief justification for this choice over more traditional recurrent architectures (like LSTMs or GRUs) would be beneficial, perhaps just a sentence in the methodology section. It is not a major issue, as the results stand, but it would help readers understand the design choice, especially given that recurrent networks are also common in this domain.

7. PLOS authors have the option to publish the peer review history of their article (what does this mean?). If published, this will include your full peer review and any attached files.

Reviewer #1: No

Reviewer #2: No

---

## [Editor Report · Acceptance letter]

PONE-D-25-48123R1

PLOS ONE

Dear Dr. Xing,

I'm pleased to inform you that your manuscript has been deemed suitable for publication in PLOS ONE. Congratulations! Your manuscript is now being handed over to our production team.

Kind regards,

on behalf of

Dr. Guangyin Jin

Academic Editor

PLOS ONE